# Unique 4-DOF Relative Pose Estimation with Six Distances for UWB/V-SLAM-Based Devices

**DOI:** 10.3390/s19204366

**Published:** 2019-10-09

**Authors:** Francisco Molina Martel, Juri Sidorenko, Christoph Bodensteiner, Michael Arens, Urs Hugentobler

**Affiliations:** 1Fraunhofer Institute of Optronics, System Technologies and Image Exploitation IOSB, Gutleuthausstraße 1, 76275 Ettlingen, Germany; 2Institute of Astronomical and Physical Geodesy, Technical University of Munich, Arcisstrasse 21, 80333 Munich, Germany

**Keywords:** sensor fusion, relative pose estimation, relative localization, ultra-wideband (UWB), augmented reality (AR), C-SLAM

## Abstract

In this work we introduce a relative localization method that estimates the coordinate frame transformation between two devices based on distance measurements. We present a linear algorithm that calculates the relative pose in 2D or 3D with four degrees of freedom (4-DOF). This algorithm needs a minimum of five or six distance measurements, respectively, to estimate the relative pose uniquely. We use the linear algorithm in conjunction with outlier detection algorithms and as a good initial estimate for iterative least squares refinement. The proposed method outperforms other related linear methods in terms of distance measurements needed and in terms of accuracy. In comparison with a related linear algorithm in 2D, we can reduce 10% of the translation error. In contrast to the more general 6-DOF linear algorithm, our 4-DOF method reduces the minimum distances needed from ten to six and the rotation error by a factor of four at the standard deviation of our ultra-wideband (UWB) transponders. When using the same amount of measurements the orientation error and translation error are approximately reduced to a factor of ten. We validate our method with simulations and an experimental setup, where we integrate ultra-wideband (UWB) technology into simultaneous localization and mapping (SLAM)-based devices. The presented relative pose estimation method is intended for use in augmented reality applications for cooperative localization with head-mounted displays. We foresee practical use cases of this method in cooperative SLAM, where map merging is performed in the most proactive manner.

## 1. Introduction

A considerable amount of research and progress has been made in the last few decades in the field of simultaneous localization and mapping (SLAM) [1,2]. SLAM methods concurrently map the surrounding three-dimensional space and estimate the own pose (position and orientation) relative to the constructed map. Augmented reality (AR) has notably benefited from advances in this field.

Cooperative SLAM (C-SLAM) is a related important topic with special relevance in robotics and it adds an additional layer of complexity, because it requires prior knowledge of the map or the relative poses (coordinate frame transformations). The relative poses are calculated either by using common information of local maps or by meeting (rendezvous) and performing relative line-of-sight (LOS) measurements. The need to scan common areas of the environment or to meet, especially if the relative poses are not known, results in a time loss, which is unacceptable in time-critical applications such as emergency rescue operations.

We have integrated an ultra-wideband (UWB) transceiver in a SLAM-based AR test device (Microsoft HoloLens), extending its functionality. The UWB transceiver uses the time of arrival (TOA) technique to measure the distance to other UWB-transceivers. UWB technology is ideal for indoor and GPS-denied applications due to its reduced sensitivity against multipath effects. It further allows the resilient localization of several SLAM-based devices in the absence of a loop closure or under harsh conditions (low lighting, smoke, dust), where cooperative visual SLAM would fail. In this work, we address the relative localization problem of two UWB/SLAM-based devices with distance measurements only.

Our AR test device has four external cameras, an inertial measurement unit (IMU) and a TOF camera. The last two sensors allow metric localization and mapping. The IMU can estimate the gravitation vector with high accuracy [3]. We restrict the six-dimensional relative pose estimation problem to four degrees of freedom (4-DOF) using this information. This decreases the minimum number of distances needed and improves the accuracy of the 3D relative pose estimation. Our method is well suited in conjunction with outlier detection algorithms such as RANSAC, since minimizing the subset sample data considerably reduces their computational complexity.

### 1.1. Related Work

Early works in C-SLAM assume that the initial poses of the robots are known [4]. More recent publications [5,6,7] use different approaches but they all rely on common map information and/or robot encounters with LOS.

In a review of C-SLAM [8], estimating the relative poses and their respective uncertainty are considered two major problems. The importance of estimating the relative transforms is also emphasized in [9,10] and [11].

Our method can be seen as an extension of rendezvous [12]. Because we use radio frequency (RF), it is possible to use LOS and non-line-of-sight (NLOS) measurements to estimate the relative poses and we estimate it with minimal data communication. In this manner a global map can be created and loop closures can be predicted without overlaps between the local maps.

Several cooperative localization (CL) methods fuse odometry readings with different types of relative measurements using a particle filter [13], a maximum likelihood estimation (MLE) [14] or an extended Kalman filter (EKF) [15] approach in order to estimate the current poses of the robots. The authors in [16] investigate the CL problem of two UAVs equipped with inertial sensors and a camera performing relative bearing measurements only. They apply an EKF to estimate the relative position, orientation and velocity between both UAVs.

The authors in [17,18] and [19] have extensively researched the problem of determining the robot-to-robot relative pose using only range measurements and local positions. They compute an initial estimate of the relative pose transformation with non-iterative algorithms and refine the solution using weighted least squares. They present solutions for the 2D and 3D case.

The work in [18] provides non-iterative algorithms for computing the initial relative pose in 2D. The authors describe a linear algorithm to estimate the solution with five or more distance measurements. They provide experimental results using wheel odometry and adding noise to the ground truth distances measured by an external camera. However, it would be interesting to test these methods under field conditions.

In [19], a linear solution with ten distance measurements is presented for the 3D problem with six degrees of freedom. In [20], the same authors provide algorithms based on polynomial system solving for the minimal problem with six distance measurements. The presented methods can have up to 40 real solutions.

In [21] several two-dimensional localization problems are addressed: localizing three agents with inter-agent range measurements and relative bearing measurements to two anchors, and the problem of relative pose estimation (relative frame determination). From graph rigidity theory they prove for the relative frame determination problem, that in the absence of noise and for generic positions, if four or more distance measurements are available, there is one unique solution. To solve these problems the authors propose a sum of squares (SOS) relaxation approach. They compare their solution with the two-dimensional relative pose determination in [18]. This non-linear method results in a lower error, however the linear method is approximately a factor 1000 faster and is accordingly very suitable for real-time computations.

Prorok and Martinoli [22] use a particle filter to fuse TDOA measurements with relative range and bearing measurements. They use multimodal probability distributions to model both LOS and NLOS UWB distance measurements. An experimental setup with a group of 10 robots is shown. Each robot is equipped with wheel odometry, a relative range and bearing module and an UWB emitter which communicates with four UWB base stations.

The authors in [23] address a 2D cooperative localization problem using an EKF in order to account for the displacement noise. With the assumption of known orientation, they provide a method to localize a device with an embedded IMU using distances to a single UWB beacon. The authors also introduce a relative localization (RL) method based on relative distance measurements between the devices. They use the first described method with a static beacon for the initialization of this RL method. The authors show the results of simulations and an experimental setup based on three unmanned aerial vehicles (UAVs), where one of them is hovering in the air, acting as a base station.

To the best of our knowledge, we have not found any experiments in the literature that integrate RF in visual-SLAM devices to estimate the relative poses without the use of external beacons.

### 1.2. Contributions

We present a relative localization method based on distance measurements between two UWB/SLAM-based devices and a novel linear algorithm for estimating the unique 4-DOF relative pose with a minimum of six distance measurements. In comparison, the linear algorithm proposed in [19] needs ten distances and a minimal solver obtains up to 40 solutions for the more general 6-DOF case [20].

The presented method can be also applied for the 2D case and presents an alternative linear method to [18], since we reduce 10% of the translation error. It estimates the relative pose uniquely between two devices with a minimum of five distances for the 2D case.

Both of the proposed 2D and 3D linear algorithms support more than the minimum required distance measurements. Using these algorithms in an outlier-free overdetermined system provides very accurate initial estimates for iterative refinement, reducing both the number of iterations and the probability of converging to a local minima.

An experimental analysis with data measured by our UWB/visual (V)-SLAM-based systems validates the effectiveness of our method.

### 1.3. Outline

Section 2 describes the distance-based relative pose estimation problem and the proposed linear closed form solution in 2D and 3D for the special case of 4-DOF. Section 3 presents simulation results for both cases and compares the error with other related linear methods [18,19]. Section 4 describes the experimental setup and shows the results with real data. Section 5 concludes with an interpretation of the achieved results.

## 2. Methods

### 2.1. Problem Formulation and Notation

We define ${}^{k}p_{i}^{\left(j\right)}$ as the set of positions, where the positions are indexed with *i*, measured by device *j* and expressed in coordinate frame *k*. Figure 1 illustrates the geometric problem in 2D. Each SLAM-based device will measure a set of positions $p_{i}^{\left(1\right)}$ and $p_{i}^{\left(2\right)}$. If not stated otherwise, they are expressed in their own coordinate systems: Coordinate frame {1} for device 1 and coordinate frame {2} for device 2. While moving, they perform distance measurements $d_{i}$ between each other. The goal is to find the relative pose between both by estimating the transform that converts the positions measured by the second device $p_{i}^{\left(2\right)}$ into the positions ${}^{1}p_{i}^{\left(2\right)}$ expresed in coordinate frame {1} of the first device. The transform is given by a translation $\vec{\varepsilon }$ and a rotation angle α. Assuming that both coordinate system origins lie at the position of the first distance measurement, we see that the Euclidian norm of the translation ${∥\vec{\varepsilon }∥}_{2}$ is equal to this first distance. This fact has been exploited in all mentioned related works. Our method can be decoupled from any related assumptions or constraints. We analyse the effect of fixing the norm of the translation to the first distance measurement in Section 3.

We formulate solutions to the problem in 2D and 3D with 4-DOF using the vector notation in Table 1. In the 2D case, we only use the *x*- and *y*-axis. In 3D, we assume that the gravity vector is known, since most mobile devices integrate an inertial measurement unit. $R_{\alpha }$ is the rotation matrix around the vertical axis with angle α.

### 2.2. Cooperative Localization in 2D

#### 2.2.1. Solution of an Overdetermined System

We generate a system of linear equations following Equation (Equation 1) with $d_{i}$ being the measured distances and $p_{i}^{\left(1\right)}$, $p_{i}^{\left(2\right)}$ the measured positions of each device.
(1)$$d_{i}^{2}={∥p_{i}^{\left(1\right)}-{}^{1}p_{i}^{\left(2\right)}∥}_{2}^{2}={\left(A_{i}-X_{i}\right)}^{2}+{\left(B_{i}-Y_{i}\right)}^{2},$$
with
(2)$$X_{i}=x_{i}\cos\alpha -y_{i}\sin\alpha +u$$
(3)$$Y_{i}=x_{i}\sin\alpha +y_{i}\cos\alpha +v.$$

Inserting Equations (Equation 2) and (Equation 3) into Equation (Equation 1) leads to the following system of equations: $b_{i}=-2A_{i}u-2B_{i}v+\left(-2A_{i}x_{i}-2B_{i}y_{i}\right)\cos\alpha +$
(4)$$\left(2A_{i}y_{i}-2B_{i}x_{i}\right)\sin\alpha +2x_{i}L_{1}+2y_{i}L_{2}+L_{3},$$
with $b_{i}:=d_{i}^{2}-A_{i}^{2}-B_{i}^{2}-x_{i}^{2}-y_{i}^{2}$ being known. The auxiliary variables $L_{1}$, $L_{2}$ and $L_{3}$ substitute the non-linear terms.
(5)$$L_{1}:=u\cos\alpha +v\sin\alpha$$
(6)$$L_{2}:=v\cos\alpha -u\sin\alpha$$
(7)$$L_{3}:=u^{2}+v^{2}.$$

We formulate Equation (Equation 4) in the linear matrix form
(8)Mx=b,
where M is the coefficient matrix and b is the column vector of the terms $b_{i}$. The solution vector $x\in R^{7}$ is defined as:
$$x:={\left(\begin{array}{ccccccc} u & v & \cos\alpha & \sin\alpha & L_{1} & L_{2} & L_{3} \end{array}\right)}^{T}.$$


Note that we do not make any assumptions of the translation length ${∥\vec{\varepsilon }∥}_{2}$ while calculating the solution. This allows us to shift the origin of the second coordinate system previous to relative pose calculation. We lay it at the center of mass (mean) of the positions $p_{i}^{\left(2\right)}$ measured by the second device. Given the calculated rotation and the translation to the center of mass of the second coordinate system, we can calculate the translation vector $\vec{\varepsilon }$. Originating from $p_{1}^{\left(1\right)}$, the translation vector should ideally have the length of the first distance measurement. We optionally constrain the Euclidian norm of the translation with ${∥\vec{\varepsilon }∥}_{2}=d_{1}$.

#### 2.2.2. Unique Solution with Five Distance Measurements

We cannot solve Equation (Equation 8) directly with five distance measurements since $rank\left(M\right)<=5<dim\left(R^{7}\right)$. We calculate the nullspace of the augmented matrix $H=\left[\begin{array}{cc} M & -b \end{array}\right]$ with M and b from Equation (Equation 8). The nullspace ker(H) is the solution subspace where the homogeneous Equation (Equation 9) holds, where $\hat{x}:={\left[\begin{array}{cc} x^{T} & 1 \end{array}\right]}^{T}$.

(9)$$H\hat{x}=0$$

Assuming that the rows in H are linearly independent, then rank(H)=5. Due to the rank-nullity theorem we can state that the dimension of the nullspace $Nullity\left(H\right)=dim\left(R^{8}\right)-rank\left(H\right)=3$. The three vectors r, s and t span the nullspace so that

(10)$$\hat{x}=\lambda_{1}r_{+}\lambda_{2}s+\lambda_{3}t.$$

From Equation (Equation 10) and the definition of $\hat{x}$ it is evident that
(11)$$\lambda_{1}r_{8+}\lambda_{2}s_{8}+\lambda_{3}t_{8}=1.$$
where $r_{8}$, $s_{8}$ and $t_{8}$ are the eighth components of the respective vectors. We use Equation (Equation 11) to eliminate $\lambda_{3}$ and use Equations (Equation 5), (Equation 6), (Equation 7), (Equation 12) and (Equation 13) as constraints,
(12)$$\cos^{2}\alpha +\sin^{2}\alpha =1$$
(13)$$L_{1}^{2}+L_{2}^{2}=u^{2}+v^{2},$$
to calculate the parameters $\lambda_{1}$, $\lambda_{2}$ and $\lambda_{3}$ using an approach similar to [18]. We substitute the corresponding elements from $\hat{x}$ in Equation (Equation 10) into the constraint equations. We then reformulate these equations in a linear matrix form (Equation 8) with the unknown vector
$$x:={\left[\begin{array}{ccccc} \lambda_{1} & \lambda_{2} & \lambda_{1}\lambda_{2} & \lambda_{1}^{2} & \lambda_{2}^{2} \end{array}\right]}^{T},$$
in order to determine these parameters uniquely.

With known parameters $\lambda_{1}$, $\lambda_{2}$ and $\lambda_{3}$ we apply Equation (Equation 10) to obtain the solution $\hat{x}$. We finally use the first distance measurement as the Euclidian norm of the translation.

### 2.3. Cooperative Localization in 3D

#### 2.3.1. Solution of an Overdetermined System

We assume that our test device is sufficiently precise for a correct vertical axis alignment with a common gravity vector and a common metric scale of the two different coordinate frames. The coordinate transformation is restricted to 4-DOF with a single elementary rotation with angle α around the vertical axis. The vector of unknowns has now an additional element *w* for the vertical component of the translation.

$$x:={\left(\begin{array}{cccccccc} u & v & w & \cos\alpha & \sin\alpha & L_{1} & L_{2} & L_{3} \end{array}\right)}^{T}$$

Following the same procedure as in Equation (Equation 1) we arrive at the 3D linear system of equations
(14)$$b_{i}=\beta_{i1}u+\beta_{i2}v+\beta_{i3}w+\beta_{i4}\cos\alpha +\beta_{i5}\sin\alpha +\beta_{i6}L_{1}+\beta_{i7}L_{2}+L_{3},$$
where $b_{i}:=d_{i}^{2}-A_{i}^{2}-B_{i}^{2}-C_{i}^{2}-x_{i}^{2}-y_{i}^{2}-z_{i}^{2}+2C_{i}z_{i}$ and the respective derived coefficients $\beta_{ij}$ are defined as $\beta_{i1}:=-2A_{i}$, $\beta_{i2}:=-2B_{i}$, $\beta_{i3}:=2z_{i}-2C_{i}$, $\beta_{i4}:=-2A_{i}x_{i}-2B_{i}y_{i}$, $\beta_{i5}:=2A_{i}y_{i}-2B_{i}x_{i}$, $\beta_{i6}:=2x_{i}$ and $\beta_{i7}:=2y_{i}$.

The auxiliary variable $L_{3}$ is now given by

(15)$$L_{3}:=u^{2}+v^{2}+w^{2}.$$

#### 2.3.2. Unique Solution with Six Distance Measurements

We use the same approach explained previously in the 2D case and calculate the nullspace of the matrix H. If the rows are linearly independent, rank(H)=6 and the dimension of the nullspace stays the same as in Section 2.2.2. The only constraints that change are $L_{3}$ with Equation (Equation 15) and Equation (Equation 11) is replaced by

(16)$$\lambda_{1}r_{9+}\lambda_{2}s_{9}+\lambda_{3}t_{9}=1.$$

## 3. Simulation Results

### 3.1. Cooperative Localization in 2D

We implement a MATLAB simulation in order to evaluate the different implemented methods. In this simulation, two devices move randomly for one to two meters between consecutive positions. At each new position they perform a distance measurement which is affected by Gaussian noise. We assume precise measurements of the local positions given by SLAM. The illustrated results are based on the average of 5000 trials for each setting on the horizontal axis. Figure 2 shows the mean absolute error of the translation for different numbers of distance measurements *N*, with the standard deviation of our UWB-based distance measurement noise $\sigma_{d}$ = 2 cm. As expected, the error decays with an increasing number of measurements. Note that for seven distances the error increases given that we solve directly for the solution with Equation (Equation 8) for N≥7. We can observe that fixing the Euclidean norm of the translation ${∥\vec{\varepsilon }∥}_{2}$ to the first distance measurement $d_{1}$ reduces the error of the translation when the number of measured distances is small. For a larger set of distances, the error induced by fixing the translation length outweighs the additional information provided by this constraint.

Applying the method described in Section 2.2.2 with five distances and fixing the translation length to the first distance measurement delivers the same result as in [18]. As our method is decoupled from the first distance measurement, we calculate the solution at the center of mass of the second coordinate system. This reduces approximately 10% of the translation estimate error. In Figure 3, we illustrate the mean absolute errors of the 2D relative pose estimation as a function of the distance measurement noise. In Figure 4, the errors are given as a function of the displacement noise. A two-dimensional Gaussian noise is added to each new movement in between distance measurements. The standard deviation of the distance measurement noise is $\sigma_{d}$ = 2 cm. We can observe the performance limitation induced by the distance measurement noise, when the displacement noise is small. We remark a similar improvement of the translation error in Figure 3b and in Figure 4b, when taking displacement noise into account.

### 3.2. Cooperative Localization in 3D

We implement a similar simulation to the previous subsection including the three-dimensional movement. We put our method into perspective with [19], which handles the more general case of relative pose estimation with 6-DOF. This linear method has a total of six unknowns and requires a minimum of ten distances. Our method only takes the yaw rotation under consideration and needs a minimum of six distances. In Figure 5, we compare the mean absolute error of the calculated translation and yaw rotation of both methods as a function of the distance measurement noise. For a fair comparison, we always apply the first distance measurement as the fixed norm of the translation. This improves the accuracy of the algorithms for small sets as shown previously. We observe that our 4-DOF method outperforms the linear 6-DOF algorithm when both using the minimum required distances. Our method shows a reduced rotation error by a factor of four at the measured distance standard deviation $\sigma_{d}$ = 2 cm of our UWB transponders. The translation error is also reduced when the standard deviation of the distance measurements is kept low. When using ten distances, our method clearly improves accuracy, reducing the orientation error by a factor of 17 and the translation error by a factor of seven at the measured distance standard deviation $\sigma_{d}$ = 2 cm of our UWB transponders.

Similarly to the previous section, we show the mean absolute error as a function of the displacement noise in Figure 6. At each new position of each device, a Gaussian three-dimensional noise with standard deviation $\sigma_{p}$ is added. Thus both devices drift away from their ground truth positions. Then again, we use the measured distance standard deviation $\sigma_{d}$ = 2 cm.

We also show the 3D error dependency of the number of measurements and the effect of fixing the length of the distance in Figure 7 for the sake of completeness.

#### Tilt Error Effect on the 4-DOF Relative Pose Estimation

Motivated by the limited accuracy that noisy one-dimensional range measurements provide to estimate the six-dimensional relative pose, we use the IMU sensor information in order to reduce the problem to four dimensions. IMU sensors are ubiquitous today and can be used to estimate the gravity vector using long term accelerometer information and the angular position, which is mainly calculated from the gyroscope data. While the gyroscope data suffers from drift over time, the gravity vector can be estimated with a tilt error well below 5° [3]. It is worth noting that the gyroscope drift can be easily mitigated using additional sensors, such as a camera in conjunction with image processing algorithms.

To study the effect of the the gravity vector tilt error on the 4-DOF relative pose estimation performance, we perform a simulation with tilted coordinate frames and compare the magnitude of the error with the related 6-DOF linear method. In Figure 8, we show the mean absolute error of the relative pose estimation as a function of the tilt error of the gravity vector. As the tilt error increases, the performance of the 4-DOF relative pose estimation degrades. The 6-DOF method implicitly estimates the relative tilt between the coordinate frames, thus its performance is barely affected. For precise range measurements and an inaccurate estimate of the gravity vector, it may be preferable to compute the full 6-DOF relative pose. The range precision of UWB however, lies in between centimeter and decimeter level. Figure 8 indicates that a reduction to 4-DOF yields a better relative pose estimation using UWB, even when the gravity vector is estimated with an IMU only.

## 4. Experimental Results

We use our test devices to perform an experimental validation of our method. Two SLAM-based AR devices integrate each an UWB transceiver (DW1000 IC evaluation board) and are shown in Figure 9. The devices take a random path in all three dimensions inside of our office, which measures 8 m in width, 4.1 m in length and 3.5 m in height. The length of the path taken in this experiment for each device is greater than 40 m. Approximately 4500 distances are measured in LOS conditions while both systems are moving. The local position data of both devices and respective distance measurements are sent to a server where the relative pose is calculated.

We take random samples out of the data measured in this experiment and perform the proposed 4-DOF linear method described in Section 2.3. The randomly selected local position data of each device lies between one and two m apart of the previous picked position, imitating the geometrical conditions of the simulation in Section 3.2. We perform 5000 calculations with random subsets of six and ten distances of the measured data. Unfavorable configurations are detected by checking the rank of M in Equation (Equation 8). The ground thruth relative pose transform is estimated using MeshLab’s iterative closest point (ICP) algorithm on both local maps.

We illustrate this process in Figure 10 by showing a random picked subset of six distance measurements from the experimental data and the respective path extract of each device. In Figure 10a, the ground truth path of each device is shown from a top view, where the random picked up positions $p_{i}^{\left(1\right)}$ and ${}^{1}p_{i}^{\left(2\right)}$ and the distance measurements $d_{i}$ are displayed by markers in the paths and black dashed lines, respectively. We show the estimated relative localization with these six distances in Figure 10b.

The resulting mean absolute errors are listed in Table 2. Overall, we observe an improvement in accuracy over the linear 6-DOF method [19] in our experimental validation. When using the same number of distances, the relative pose estimation is improved by approximately a factor of ten.

The server is able to visualize the aligment of local maps. Figure 11 shows an example of map merging performed with distance-based relative pose estimation. Two local maps (green and violet) are correctly overlaid after applying the estimated transform. They show the office where the experiments were performed. One of the devices had previously visited the hallway of our institute building (illustrated in violet colour).

## 5. Conclusions

We presented a linear method for solving the relative pose between two SLAM-based devices with distance measurements only. The method is applicable to 2D and to 3D problems for the special case of 4-DOF. The 2D algorithm represents an alternate method to the linear solution proposed in [18].

The 3D algorithm calculates the 4-DOF relative pose with only six distances, in contrast to 10 distances in the 6-DOF method presented in [19]. Using six distance measurements with our method results in a more accurate solution as the 6-DOF linear method with 10 distances. Our proposed method has a rotational mean absolute error that is always smaller and the translation error is also smaller if the standard deviation of the distance measurements is kept in the centimeter level or below. This holds true in our application, as the measured standard deviation of our UWB devices using time of arrival is $\sigma_{d}$ ≈ 2 cm. When using the same amount of distances, our 3D method is clearly superior. The proposed linear method imposes the advantage over minimal 3D methods [20] that it can calculate the unique solution directly with the same amount of distances and avoids the post processing phase needed to discard all inconsistent solutions.

The proposed method decouples dependency of a fixed translation length to the problem and we show improved results for larger sets. Decoupling also allows to estimate the frame transformation to the center of mass of the second coordinate system, improving the results of the translation estimate.

Both 2D and 3D proposed algorithms can be used with any number of measurements larger than the minimal unique solution, providing a very accurate estimate of the relative pose solution for larger outlier-free sets. This estimate lies near the global minimum and we further refine it with an iterative least squares method.

We have implemented simulations of the proposed methods and performed first experiments with two UWB/SLAM-based AR test devices for cooperative localization and mapping.

## 6. Patents

Patent pending status. 

## Figures and Tables

**Figure 1 sensors-19-04366-f001:**
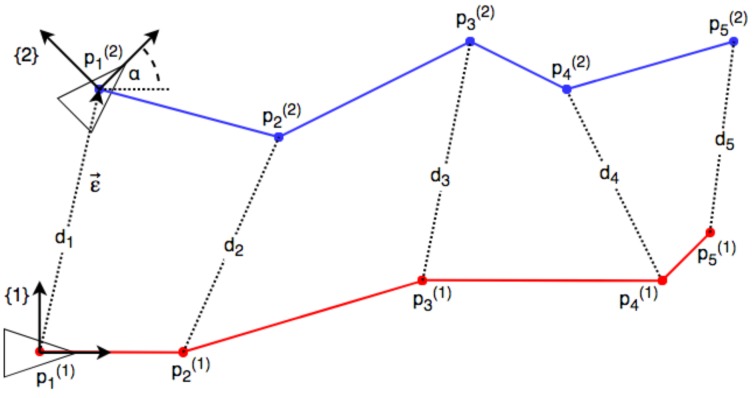
Illustration of the 2D relative pose problem.

**Figure 2 sensors-19-04366-f002:**
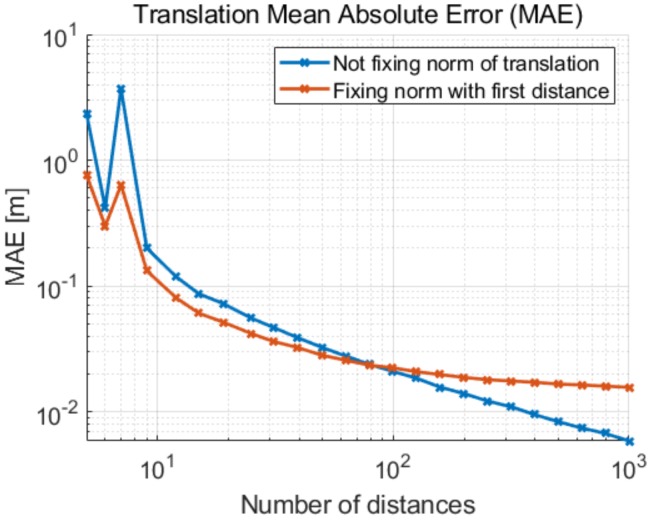
Mean absolute error (MAE) of the translation in 2D as a function of the number of distance measurements with standard deviation $\sigma_{d}$ = 2 cm.

**Figure 3 sensors-19-04366-f003:**
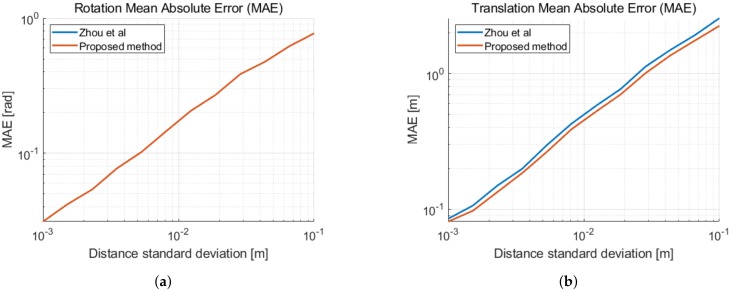
Mean absolute error (MAE) as a function of the distance standard deviation $\sigma_{d}$ in 2D. (**a**) MAE of the rotation with respect to the ground truth. (**b**) MAE of the translation with respect to the
ground truth.

**Figure 4 sensors-19-04366-f004:**
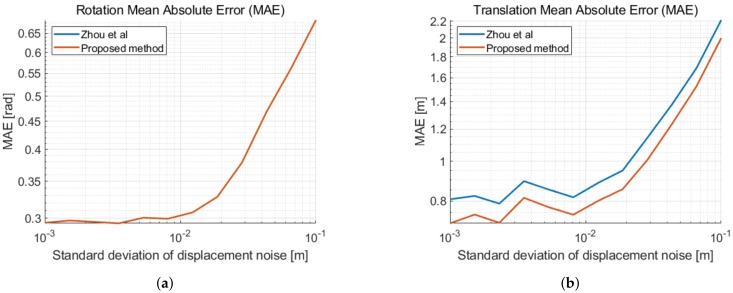
Mean absolute error (MAE) as a function of the standard deviation of the displacement noise $\sigma_{p}$ and a distance standard deviation $\sigma_{d}$ = 2 cm. (**a**) MAE of the rotation with respect to the ground truth. (**b**) MAE of the translation with respect to the ground truth.

**Figure 5 sensors-19-04366-f005:**
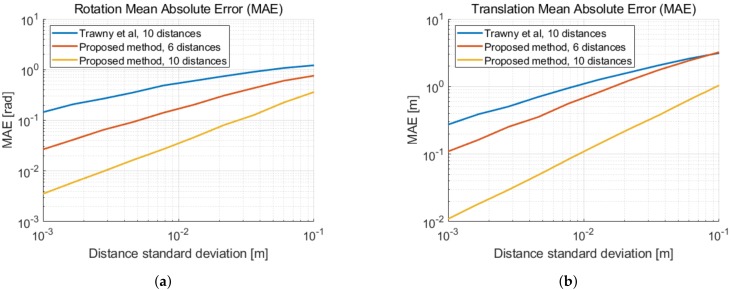
Mean absolute error (MAE) as a function of the distance standard deviation $\sigma_{d}$ in 3D. (**a**) MAE of the yaw rotation with respect to the ground truth. (**b**) MAE of the translation with respect to the ground truth.

**Figure 6 sensors-19-04366-f006:**
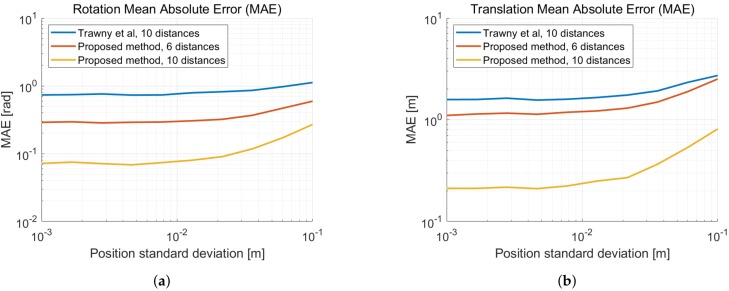
Mean absolute error (MAE) as a function of the standard deviation of the displacement noise $\sigma_{p}$ and a distance standard deviation $\sigma_{d}$ = 2 cm. (**a**) MAE of the rotation with respect to the ground truth. (**b**) MAE of the translation with respect to the ground truth.

**Figure 7 sensors-19-04366-f007:**
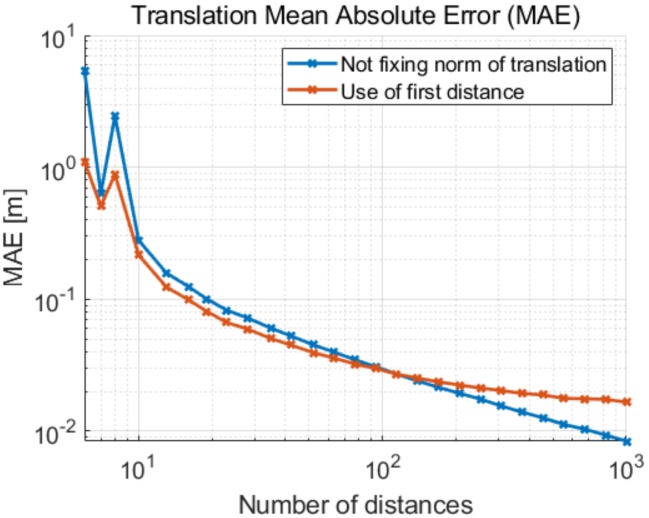
Mean absolute error (MAE) of the translation in 3D as a function of the number of distance measurements with standard deviation $\sigma_{d}$ = 2 cm.

**Figure 8 sensors-19-04366-f008:**
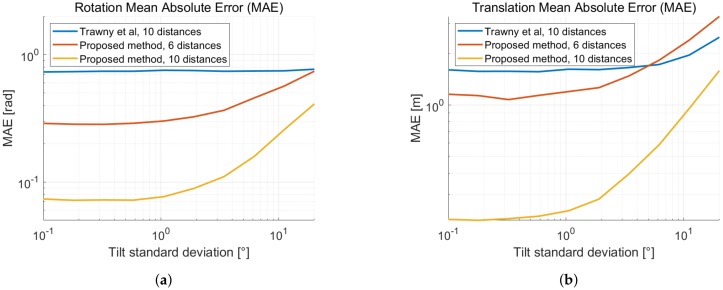
Mean absolute error (MAE) as a function of the standard deviation of the gravity vector tilt error and with distance standard deviation $\sigma_{d}$ = 2 cm. (**a**) MAE of the yaw rotation with respect to the ground truth. (**b**) MAE of the translation with respect to the ground truth.

**Figure 9 sensors-19-04366-f009:**
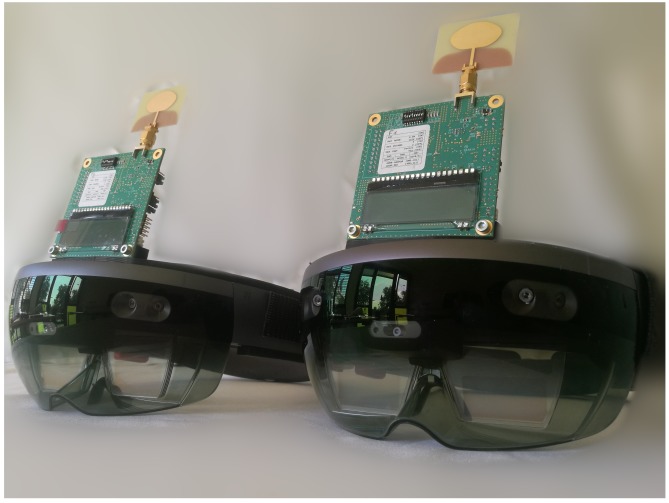
Simultaneous localization and mapping (SLAM)-based augmented reality (AR) devices with integrated ultra-wideband (UWB) transceivers.

**Figure 10 sensors-19-04366-f010:**
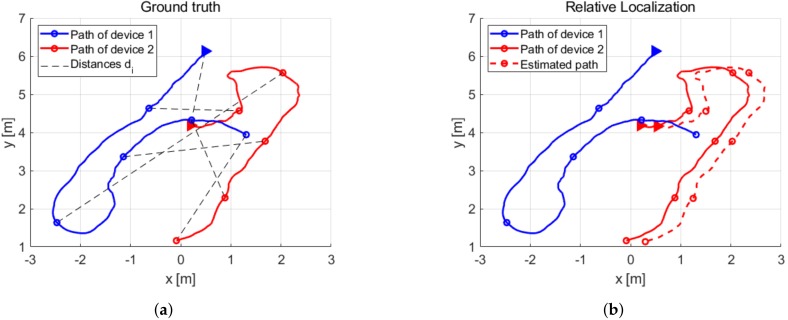
Several subsets of the measured data are selected in order to validate our method. (**a**) Ground truth paths of each device and the selected distance measurements. (**b**) Comparisson of the estimated path of device 2 with the ground truth.

**Figure 11 sensors-19-04366-f011:**
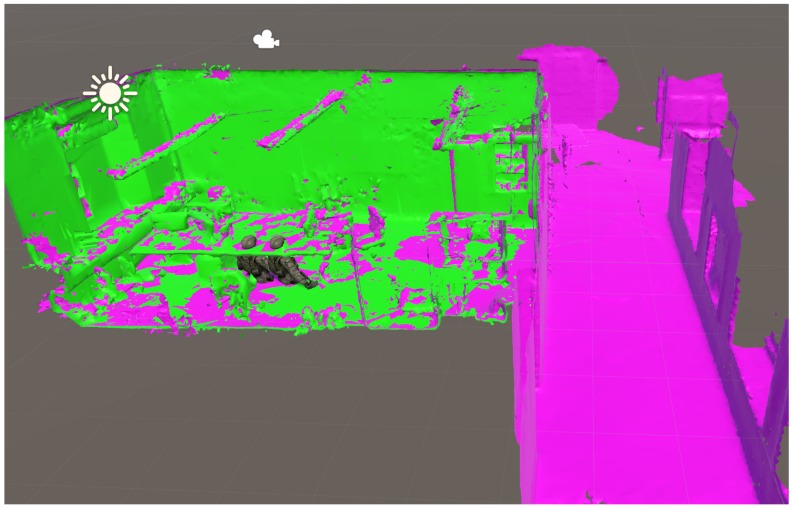
Map merging with UWB/visual (V)-SLAM-based relative pose estimation.

**Table 1 sensors-19-04366-t001:** Vector notation.

Vector/Points	*x*-Axis	*y*-Axis	*z*-Axis
$p_{i}^{\left(1\right)}$	$A_{i}$	$B_{i}$	$C_{i}$
$p_{i}^{\left(2\right)}$	$x_{i}$	$y_{i}$	$z_{i}$
${}^{1}p_{i}^{\left(2\right)}=R_{\alpha }p_{i}^{\left(2\right)}+\vec{\varepsilon }$	$X_{i}$	$Y_{i}$	$Z_{i}$
$\vec{\varepsilon }$	*u*	*v*	*w*

**Table 2 sensors-19-04366-t002:** Experimental mean absolute errors.

Method	Translation MAE	Rotation MAE
Trawny et al, 10 distances	3.42 m	27.2°
Own method, 6 distances	1.0 m	15.94°
Own method, 10 distances	0.26 m	2.68°

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
