# Peer review of "Unique 4-DOF Relative Pose Estimation with Six Distances for UWB/V-SLAM-Based Devices"

_sensors, 2019, doi:10.3390/s19204366_

Round 1

Reviewer 1 Report

Dear Authors,

I have reviewed the paper entitled “Unique 4-DOF relative pose estimation with six distances for UWB/V-SLAM-based devices”
I think it needs a minor revision. 
Please see below some remarks, 
Kind regards.

In my opinion, the literature review needs to be improved. The authors have not reviewed the current solutions used in science and practice. I don't want to quote names, but I recommend reviewing the Web of Science and Scopus database with a keyword such as: photogrametry; SLAM .

(especially from Europe, outside Germany).
I miss the reference to the earlier works of the authors of the manuscript I reviewed. You may want to consider expanding the literature review.

Please complete the:
- in DOI references (if available),
- in affiliation, the country of origin of the authors of this manuscript (Germany).

Author Response

Dear Reviewer,

Regarding the literature review, we were cautious to put too much emphasis in SLAM or photogrametry literature, since in our opinion, it is not central to the research of this work. SLAM is more a basis on which we work on to perform sensor fusion and cooperative localization. Furthermore, we use SLAM and C-SLAM as a motivation for our research to be done. Nevertheless, we added a book [1] into the literature review. Several related works, which in our opinion are closer to this publication in particular (cooperative localization, ultra-wideband and relative pose estimation), are listed in the literature review section.

DOI references and affiliation have been added/changed as requested.

Kind regards,

Francisco Molina Martel

Reviewer 2 Report

Overall, the paper is of high technical quality. The method is well explained, the experiments clearly defined and the results for the most part convincingly presented. The literature review seems to be comprehensive and the methods highlighted as benchmark seem appropriate.

However, there is one issue I feel the paper should address in more detail. It stands to reason that a solution specifically designed for 4DOF will outperform a more general 6DOF method on a 4DOF problem. However, to which extent is this advantage retained when the extra information used (the measurement of the gravity vector by the inertial device) is noisy? The real-world experiment seems to suggest that although the advantage is reduced (improvement factor reduced to 13 from 17) it is still very significant, but perhaps the question can be easily answered with more simulation results?

I think the case where the normal of the local motion plane of the robot/vehicle is known only approximately (i.e. horizontal) is very common in robots and automotive applications alike and this unanswered research question is therefore relevant. 

A similar remark concerns the general assumption that the local poses obtained from single-device SLAM are precise. Is it representative to model only distance measurement noise, and not noise on the self-estimated pose of each device? Measurement noise induced by low-grade IMU may play a larger role than UWB distance measurement noise.

Author Response

Dear Reviewer,

Thank you very much for your valuable suggestions. As you proposed, we performed a new simulation with noisy gravitation vector measurement, in order to compare the 4-DOF performance decrease with increasing tilt error against the full relative pose estimation. To present this result, a new subsection has been written.

Your question regarding the assumption that the poses are precise is good and at the same time not a trivial one. In the related works of relative pose estimation [18, 19], the same assumption has been done for the linear solution. There, a weighted least squares refinement is proposed in order to compensate for the drift effect on the solution. Modeling IMU drift error is rather straightforward. In the context of SLAM, it is rather complicated, since position uncertainty can change over time (e.g. due to loop closures). Working with Covariance Matrices could be a possibility, if available.

Trying to reduce the position uncertainty with relative range measurements is also a valid research direction, but suffers from difficulties, specially if no anchors are used, since a relative pose is calculated from rather noisy one-dimensional measurements, while SLAM localization is very accurate (at least at small & medium scales).

Nevertheless, we also added a 3D relative pose simulation as a function of drift in position.

Kind regards,
Francisco Molina Martel